# Towards Multiplier-Free Transformers with Stochastic Attention

## Abstract

In standard attention, a substantial fraction of compute comes from multiplying softmax weights by high-precision value vectors — even in ternary models such as BitNet, which remove multipliers elsewhere. We present Stochastic Additive No-mulT Attention (SANTA), a drop-in inference-time replacement that eliminates these value-stage multiplications. For each query, SANTA samples from the post-softmax distribution, gathers and sums selected values, and applies a single bit-shift normalization, with no expensive multipliers on the value path. SANTA's compute scales as $O(n_{queries} \cdot S \cdot d_k)$: linear in the number of queries during prefill and linear in the sample budget $S$ during decode, while exhibiting sparse, index-based memory access. SANTA is an unbiased Monte Carlo estimator of dense attention and is orthogonal to upstream efficiency techniques (ternary quantization, low-rank kernels, sparsity, pruning). Combined with existing 1-bit/ternary quantizers, SANTA moves Transformers toward fully multiplier-free, energy-efficient inference.

## 1 Introduction

The transformer architecture has become ubiquitous (Vaswani et al., 2017). State-of-the-art LLMs such as GPT (OpenAI, 2024), DeepSeek (DeepSeek-AI, 2025), Llama (Touvron et al., 2023), and Gemini (Team et al., 2023) are already practical for deployment as virtual assistants. These models come with substantial compute and memory demands, particularly in the attention mechanism, where the KV-cache scales linearly and the attention computation scales quadratically with sequence length. A growing body of work seeks to reduce inference costs, especially for edge devices and memory-limited systems.

A parallel line of work focuses on quantization, with 1-bit architectures such as BitNet (Wang et al., 2023; Ma et al., 2025; 2024) representing model parameters as $-1$, 0 and $+1$ weights outside of token embeddings. This replaces matrix multiplications with cheap, energy-efficient addition operations. However, attention matmuls between queries ($Q$), keys ($K$), and values ($V$) still rely on dense matrix multiplications in higher (8-16 bit) precision, which remain a bottleneck.

We propose **S**tochastic **A**dditive **N**o-mul**T** **A**ttention (SANTA), an inference-time method that replaces the $V$ multiplication with sampling, indexing, addition, and a bit-shift based normalization. SANTA eliminates all value stage multiplications and is complementary to existing improvements in the score stage $QK^\top$. SANTA makes sparse memory accesses to only a few rows of the value matrix and produces an unbiased estimate of dense attention. Along with 1-bit quantization, SANTA suggests a practical path toward multiplier-free transformer inference.

## 2 Related work

### 2.1 Quantization

Recent years have observed progressively more extreme quantization of model weights. Works such as QLoRA (Dettmers et al., 2023) represent model weights with as low as 4-bit precision in

---

**Code:** `https://anonymous.4open.science/r/SANTA-718E`

fine-tuning LLMs. Post-training quantization has demonstrated significant promise, storing weights in >100 billion parameter GPT models in as few as 3-4 bits (Frantar et al., 2022). Recently, transformers such as BitNet feature model parameters as ternary (-1, 0, and +1) weights with minimal accuracy loss (Wang et al., 2023; Ma et al., 2025; 2024). This turns dense matrix multiplications in projection and feedforward layers into cheap addition operations. However, the multiplication of query, key, and value matrices is left in high, $\geq$ 8-bit precision.

## 2.2 EFFICIENT ATTENTION

Implementations that keep exact attention but fuse/tile the kernel for better memory locality include FlashAttention (Dao, 2023; Dao et al., 2022) and Slim Attention (Graef & Wasielewski, 2025). They still fully evaluate $QK^\mathsf{T}$, but eliminate redundant reads/writes. Sparse-pattern variants such as Longformer and BigBird (Child et al., 2019; Zaheer et al., 2020; Beltagy et al., 2020) compute only a subset of query–key dot products, while low-rank approximations like Linformer (Wang et al., 2020) project $Q$ and/or $K$ to lower dimensions before the multiply. Recently, NoMAD-Attention removes most multiplications in the score stage by product-quantizing keys and converting each dot product into SIMD look-ups and integer adds on CPUs (Zhang et al., 2024). These techniques target the $QK^\mathsf{T}$ computation. By contrast, SANTA leaves the score stage untouched and instead targets the softmax–by-$V$ multiply. SANTA is complementary and orthogonal to $QK^\mathsf{T}$ techniques.

## 2.3 TOP-K AND KV-CACHE SPARSITY

Many attention implementations leverage the observation that only a few keys with high attention scores carry the majority of information, while unimportant keys can be dropped. For instance, top-k attention (Gupta et al., 2021) selects the k highest-score keys while discarding the rest, saving global memory accesses to rows of the $V$ matrix.

Recent work, including Quest (Tang et al., 2024) and H$_2$O (Zhang et al., 2023), avoids scoring all keys by restricting the KV subset scored before softmax. Since these KV-cache methods still perform a softmax-$V$ multiplication, they are orthogonal to SANTA, which replaces the softmax-$V$ multiplication.

Top-k attention occupies a similar architectural slot as SANTA, thus top-k is SANTA's natural competitor. While top-k attention has shown impressive performance, it is biased and lacks theoretical guarantees. Top-k's bias manifests when the attention distribution becomes a long-tailed distribution, which can occur in practice. Recent work (Chen et al., 2024) empirically shows that tasks such as common word extraction see top-k performance degrade significantly, while unbiased Monte-Carlo attention approximations retain high accuracy.

## 2.4 MONTE CARLO ATTENTION

Monte Carlo estimators have been proposed for efficient attention. Some proposals apply randomized linear algebra for partial matrix multiplications, saving FLOPs (Kim & Ko, 2022). Locality sensitive hashing (Chen et al., 2024; Kitaev et al., 2020) builds hash tables for query-key comparisons, representing a stochastic process which provides theoretical guarantees and unbiased attention estimators. Compared to existing work, SANTA is the first to propose Monte Carlo attention in a fashion that removes multiplication by $V$ after the softmax step.

## 3 STOCHASTIC ADDITIVE NO-MULT ATTENTION (SANTA)

### 3.1 ALGORITHM AND ESTIMATOR

We consider the attention mechanism as a function of key, query, and value matrices:

$$\text{Attn}(Q, K, V) = \text{softmax}\left(\frac{QK^\mathsf{T}}{\sqrt{d_k}}\right) V = AV, \tag{1}$$

where $A \in \mathbb{R}^{n_q \times n_k}$ is the row-stochastic attention score matrix and $V \in \mathbb{R}^{n_k \times d_v}$ is the value matrix.

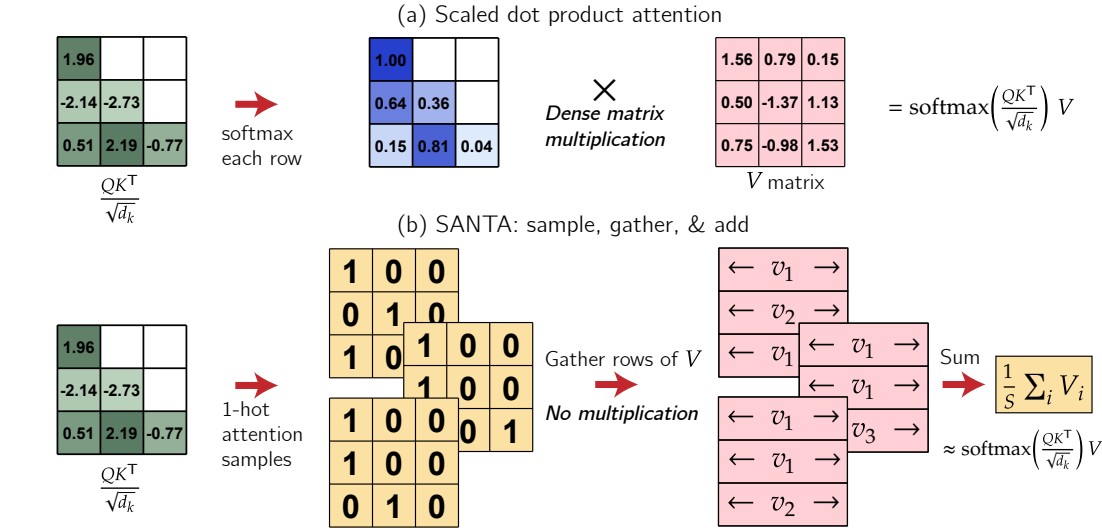

Figure 1: Scaled dot product attention (SDPA) vs. Stochastic Additive No-mulT Attention (SANTA). (a) SDPA kernels feature an expensive matrix multiplication by the value matrix $V$, requiring global memory accesses to every row of $V$. (b) SANTA eliminates multiplications following the softmax operation, using cheap additions and sparse memory accesses to sampled rows of $V$. Normalization by $S$ is a single bitshift if $S$ is a power of 2.

To approximate $AV$ without multiplications, we treat each row $A_q$ as a categorical distribution over keys and sample $S$ values i.i.d. from it. For query $q$, we fetch the corresponding $S$ rows of $V$ and average them. This yields an estimate of the attention output using only row indexing and addition.

As an illustrative example, consider $n_q = n_k = 3$. We sample $S = 3$ one-hot rows per query:

$$\tilde{A}V = \frac{1}{3}\left(\begin{bmatrix} 1 & 0 & 0 \\ 0 & 1 & 0 \\ 1 & 0 & 0 \end{bmatrix} + \begin{bmatrix} 1 & 0 & 0 \\ 1 & 0 & 0 \\ 0 & 0 & 1 \end{bmatrix} + \begin{bmatrix} 1 & 0 & 0 \\ 1 & 0 & 0 \\ 0 & 1 & 0 \end{bmatrix}\right)V, \tag{2}$$

which simplifies by distributivity of matrix multiplication over addition:

$$\tilde{A}V = \frac{1}{3}\left(\begin{bmatrix} 1 & 0 & 0 \\ 0 & 1 & 0 \\ 1 & 0 & 0 \end{bmatrix}V + \begin{bmatrix} 1 & 0 & 0 \\ 1 & 0 & 0 \\ 0 & 0 & 1 \end{bmatrix}V + \begin{bmatrix} 1 & 0 & 0 \\ 1 & 0 & 0 \\ 0 & 1 & 0 \end{bmatrix}V\right) = \frac{1}{S}(V_1 + V_2 + V_3 + \cdots), \tag{3}$$

where each $V_i$ is a matrix formed by stacking $n_q$ rows selected from $V$. Each row of $V_i$ corresponds to a single sampled key for that query. Multiplying a one-hot attention matrix with $V$ requires no arithmetic: it is a gather operation. Thus, the product $\tilde{A}V$ can be computed with $S$ indexing steps and a single vector-wise addition per query. The one-hot attention matrices in Eq. 2 and 3 are for illustrative purposes only; those matrices represent indexing operations and are *not* materialized in memory.

We now transition from this intuitive explanation to a formal description of the estimator:

**Formal description.** Let $A = \text{softmax}\left(QK^\mathsf{T}/\sqrt{d_k}\right) \in \mathbb{R}^{n_q \times n_k}$ and $V \in \mathbb{R}^{n_k \times d_v}$. For each query $q \in \{1, \ldots, n_q\}$, let $A_q$ denote its attention weight vector (a probability distribution over keys).

**Definition 3.1** (SANTA estimator). For each $q$, independently sample $i_{q,1}, \ldots, i_{q,S} \overset{\text{i.i.d.}}{\sim}$ Categorical$(A_q)$, and define

$$\widehat{V}_q = \frac{1}{S} \sum_{s=1}^{S} V_{i_{q,s}} \in \mathbb{R}^{d_v}.$$

Stacking all query outputs yields the estimate $\widehat{AV} = [\widehat{V}_1; \ldots; \widehat{V}_{n_q}] \in \mathbb{R}^{n_q \times d_v}$.

**Remark 3.2.** Choosing $S = 2^m$ allows the division to be implemented as a bit shift.

**Proposition 3.3** (Unbiasedness). $\mathbb{E}[\widehat{AV}] = AV$.

*Proof.* Fix any $q$. Since $\Pr(i_{q,s} = j) = A_{qj}$,

$$\mathbb{E}[\widehat{V}_q] = \frac{1}{S} \sum_{s=1}^{S} \sum_{j=1}^{n_k} A_{qj} V_j = \sum_{j=1}^{n_k} A_{qj} V_j = (AV)_q.$$

Linearity of expectation completes the result. $\square$

Finally, SANTA can be summarized as:

$$\tilde{A}V = \frac{1}{S} \sum_{i=1}^{S} V_i \approx AV, \tag{4}$$

where each $V_i$ is a gathered matrix of rows from $V$. Multiplications in the value stage have therefore been eliminated in favor of indexing and addition. SANTA can be viewed as an average over stochastic hard attention where each query attends to a single key but remains unbiased and tunable. Fig. 1 illustrates the difference from standard SDPA.

### 3.2 VARIANCE AND TAIL BOUNDS FOR SANTA

Throughout this subsection, fix a query index $q \in \{1, \ldots, n_q\}$. Let $A_q \in \Delta^{n_k - 1}$ denote the attention weights over keys, write $p_{qj} \triangleq A_{qj}$, and let $\mu_q \triangleq \sum_{j=1}^{n_k} p_{qj} V_j = (AV)_q$ be the exact attention output for that query. Default SANTA draws $i_{q,1}, \ldots, i_{q,S} \overset{\text{i.i.d.}}{\sim} \text{Categorical}(A_q)$ and returns

$$\widehat{V}_q = \frac{1}{S} \sum_{s=1}^{S} V_{i_{q,s}} \in \mathbb{R}^{d_v}.$$

Define centered summands $X_s \triangleq V_{i_{q,s}} - \mu_q$ and the (query-specific) covariance $\Sigma_q \triangleq \text{Cov}(V_i) = \sum_{j=1}^{n_k} p_{qj} (V_j - \mu_q)(V_j - \mu_q)^\top$.

**Proposition 3.4** (Variance scaling of SANTA). *SANTA is unbiased, and its covariance scales as* $1/S$:

$$\mathbb{E}[\widehat{V}_q] = \mu_q, \qquad \text{Cov}(\widehat{V}_q) = \frac{1}{S} \Sigma_q.$$

*Equivalently, $\mathbb{E}\left[ \|\widehat{V}_q - \mu_q\|_2^2 \right] = \frac{1}{S} \text{tr}(\Sigma_q)$.*

*Proof.* Unbiasedness was shown above. For the variance, write $\widehat{V}_q - \mu_q = \frac{1}{S} \sum_{s=1}^{S} X_s$ with $\mathbb{E}[X_s] = 0$ and $\text{Cov}(X_s) = \Sigma_q$. The $X_s$ are i.i.d., hence $\text{Cov}\left( \frac{1}{S} \sum_{s=1}^{S} X_s \right) = \frac{1}{S^2} \sum_{s=1}^{S} \text{Cov}(X_s) = \frac{1}{S} \Sigma_q$. Taking the trace gives the mean-squared error identity. $\square$

**Remark 3.5.** A dimension-free high-probability concentration bound for SANTA is deferred to Appendix A (Theorem A.1)

### 3.3 $S^2$ANTA: STRATIFIED AND SYSTEMATIC SAMPLING

S²ANTA reduces variance via stratified sampling: dividing the CDF into $S$ equal intervals and sampling once per interval. Two variants: independent (**$S^2$ANTA-ind**) with proven variance reduction, and systematic (**$S^2$ANTA-sys**) with empirically similar performance.

**Construction.** Fix a query $q$. Let $A_q = (p_{q1}, \ldots, p_{qn_k})$ be the attention weights and $F_q$ the associated CDF on $[0, 1)$. Partition $[0, 1)$ into $S$ equal intervals

$$I_m := [m/S, (m+1)/S), \qquad m = 0, \ldots, S-1.$$

Define two schemes:

- **Independent stratified ($S^2$ANTA-ind).** Draw $T_m \sim \mathrm{Unif}(I_m)$ *independently* for $m = 0, \ldots, S-1$, set $J_m := F_q^{-1}(T_m)$, and output $\widehat{V}_q^{\mathrm{ind}} := \frac{1}{S} \sum_{m=0}^{S-1} V_{J_m}$.

- **Systematic ($S^2$ANTA-sys).** Draw a single $U \sim \mathrm{Unif}([0, 1/S))$ and take thresholds $T_m := U + m/S$. With $J_m := F_q^{-1}(T_m)$, output $\widehat{V}_q^{\mathrm{sys}} := \frac{1}{S} \sum_{m=0}^{S-1} V_{J_m}$.

Both replace $S$ i.i.d. multinomial draws by one sample per equal-probability stratum and use only gathers and additions on $V$.

**Proposition 3.6** (Unbiasedness of $S^2$ANTA). *Let $\mu_q := \sum_j p_{qj} V_j = (AV)_q$. Then $\mathbb{E}[\widehat{V}_q^{\mathrm{ind}}] = \mathbb{E}[\widehat{V}_q^{\mathrm{sys}}] = \mu_q$.*

*Proof.* For $S^2$ANTA-ind,

$$\mathbb{E}[\widehat{V}_q^{\mathrm{ind}}] = \frac{1}{S} \sum_{m=0}^{S-1} \mathbb{E}[V_{F_q^{-1}(T_m)}] = \int_0^1 \left( \sum_j \mathbf{1}\{F_q(j-1) \le t < F_q(j)\} V_j \right) dt = \sum_j p_{qj} V_j = \mu_q.$$

For $S^2$ANTA-sys, condition on $U$; each $T_m$ is uniform on $I_m$ (though dependent), so the same integral argument applies after averaging over $U$; see Cochran (1977, Ch. 8) and Owen (2013). $\square$

**Remark 3.7** (Systematic vs. independent: guarantees and practice). $S^2$ANTA-ind is unbiased and admits a universal variance-dominance theorem, guaranteeing lower variance than that of i.i.d. sampling with default SANTA (see Appendix B). $S^2$ANTA-sys is also unbiased, but does *not* provide tractable variance guarantees. Empirically, we measure variance and benchmark performance comparable to $S^2$ANTA-ind.

### 3.4 THEORETICAL FLOPs AND MEMORY TRAFFIC

Table 1: Per-head FLOPs and DRAM traffic for scaled dot-product attention (SDPA), top-$k$, and SANTA. All expressions are normalized per attention head. $n_q$: queries, $n_k$: keys, $d_k$: head dimension, $k$: top-$k$ budget, $S$: sample budget.

| Stage & Variant | Adds | Mults / Divs | Reads | Writes |
|---|---|---|---|---|
| **Score stage** $(QK^{\mathsf{T}})$ | $n_q n_k (d_k - 1)$ | $n_q n_k d_k$ | $(n_q + n_k) d_k$ | $n_q n_k$ |
| **Value stage** | | | | |
| Dense SDPA | $n_q n_k d_k$ | $n_q n_k d_k$ | $n_q n_k d_k$ | $n_q d_k$ |
| Top-$k$ | $n_q k d_k$ | $n_q k d_k$ | $n_q k d_k$ | $n_q d_k$ |
| SANTA | $n_q (S-1) d_k$ | $0$ | $n_q S d_k$ | $n_q d_k$ |

We now analyze the computational cost of SANTA against SDPA and its closest competitor, top-$k$ attention, focusing on FLOPs and memory traffic in the value stage. Table 1 summarizes the theoretical costs per attention head.

While top-$k$ attention is memory-efficient and generally performs well, it introduces bias and can degrade significantly when the attention distribution is heavy-tailed. For example, recent work (Chen et al., 2024) shows that tasks such as common word extraction suffer when relevant information is not concentrated in the top-$k$ entries. In contrast, stochastic estimators like SANTA provide an unbiased estimate of full attention and retain robustness in such cases.

Table 1 summarizes theoretical FLOPs and memory traffic per forward pass for SDPA, top-$k$, and SANTA. We denote $n_q$ queries, $n_k$ keys, and $d_k$ the head dimension. $n_q$ is equal to the prompt length

in prefill and $n_q = 1$ during autoregressive generation. All expressions are normalized per attention head, and total cost scales linearly with the number of heads $H$.

For both top-k and SANTA, we define a notion of compute budget, where $k$ is the number of keys selected by top-k for each query of each attention head, while $S$ is a sample budget representing the number of samples that SANTA draws for each query of each attention head. By default, we employ uniform $S$ across model layers, though ablations suggest that $S$ can be optimized per layer (Appendix C, D).

We omit softmax computation, top-$k$ sorting overhead, and sampling from this analysis, as they are lightweight relative to the $V$ matrix multiply and highly implementation-dependent. For example, many implementations use partial sorting (Yerram et al., 2024; Xie et al., 2024) or prune the score matrix early (Zhang et al., 2023; Tang et al., 2024).

In both FLOPs and memory reads, SDPA scales as $n_q n_k$, reflecting the cost of full quadratic attention computation. Both top-k and SANTA reduce this to $n_q k$ and $n_q S$, respectively, assuming fixed budgets with $k, S \ll n_k$. This shifts the overall complexity of the value stage from quadratic to linear in sequence length, given a fixed budget $k$ or $S$.

Top-k still incurs multiplication costs while SANTA eliminates multiplies. Top-$k$ performs $n_q k d_k$ multiplications (per head), while SANTA requires only $n_q d_k$ divisions for normalization (if $S$ is chosen as a power of 2, even these few divisions become simple bit-shifts).

## 4 RESULTS AND BENCHMARKING

### 4.1 COMPARISON METHODOLOGY

**Comparing SANTA and top-k requires care, as they have different computational profiles.** We present results for sample budget $S$ equal to the top-k budget $k$ not as an iso-compute benchmark, but to demonstrate that SANTA achieves comparable or superior accuracy while using a fundamentally cheaper set of operations. At $S = k$: top-k costs $n_q k d_k$ multiplications + $n_q k d_k$ additions, whereas SANTA costs 0 multiplications + $n_q S d_k$ additions (Table 1).

A 32-bit FP multiply costs 3.7 pJ versus 0.9 pJ for addition (Horowitz, 2014). For equal budgets $k = S$, the value-stage energy consumption for the two methods are approximately: $k \times (3.7 + 0.9) = 4.6k$ pJ per element for top-k, vs. $S \times 0.9 = 0.9S$ pJ per element for SANTA. Relative to top-k, SANTA yields a $\approx 5\times$ energy reduction for value stage FLOPs. We acknowledge that these are approximate energy estimates; we aim to provide hardware-agnostic comparisons, as modern GPU hardware is optimized for matmuls and contiguous memory access, while SANTA involves adds and sparse memory access.

Regarding memory accesses: SANTA's accesses to elements of the $V$ matrix scale as $n_q S d_k$, which *in the worst case* matches top-k's $n_q k d_k$ memory accesses at $k = S$. However, since SANTA samples with replacement, the number of *unique* keys is $\leq S$. In practice, as we later quantify in Table 5, the number of unique keys sampled is $<< S$, which may offer significant caching advantages.

### 4.2 GSM8K

Table 2: GSM8K accuracy and average context length (prompt + answer). $k$: number of keys in top-$k$, $S$: SANTA sample budget. Accuracy shows 95% bootstrap confidence intervals.

| | DeepSeek-R1-Distill-Qwen-7B | | | | Llama-3.1-8B-Instruct | | | | | | | |
|---|---|---|---|---|---|---|---|---|---|---|---|---|
| | Top-$k$ | | SANTA | | Top-$k$ | | SANTA | | $S^2$ANTA-ind | | $S^2$ANTA-sys | |
| $k\|S$ | Acc. (%) | Tok. | Acc. (%) | Tok. | Acc. (%) | Tok. | Acc. (%) | Tok. | Acc. (%) | Tok. | Acc. (%) | Tok. |
| 2 | $0.23 \pm 0.15$ | 4049 | $0.00 \pm 0.00$ | 4217 | $1.21 \pm 0.32$ | 430 | $1.26 \pm 0.34$ | 1075 | $1.01 \pm 0.32$ | 1071 | $1.11 \pm 0.34$ | 1071 |
| 4 | $23.88 \pm 1.31$ | 3298 | $0.08 \pm 0.09$ | 4208 | $6.27 \pm 0.72$ | 583 | $1.57 \pm 0.37$ | 825 | $1.54 \pm 0.38$ | 611 | $1.67 \pm 0.40$ | 569 |
| 8 | $72.25 \pm 1.35$ | 2332 | $52.46 \pm 1.57$ | 2799 | $41.32 \pm 1.50$ | 549 | $1.44 \pm 0.38$ | 207 | $1.74 \pm 0.40$ | 325 | $2.10 \pm 0.45$ | 340 |
| 16 | $82.51 \pm 1.21$ | 1860 | $79.00 \pm 1.28$ | 1778 | $62.88 \pm 1.57$ | 430 | $5.51 \pm 0.72$ | 350 | $39.12 \pm 1.50$ | 352 | $44.63 \pm 1.58$ | 349 |
| 32 | $86.23 \pm 1.02$ | 1694 | $83.19 \pm 1.23$ | 1667 | $72.50 \pm 1.34$ | 384 | $38.26 \pm 1.43$ | 348 | $67.00 \pm 1.48$ | 343 | $68.59 \pm 1.42$ | 339 |
| 64 | $86.76 \pm 1.03$ | 1624 | $84.99 \pm 1.09$ | 1587 | $76.12 \pm 1.32$ | 358 | $63.63 \pm 1.49$ | 346 | $74.43 \pm 1.31$ | 343 | $\mathbf{76.42 \pm 1.34}$ | 341 |
| 128 | $87.54 \pm 1.02$ | 1626 | $85.42 \pm 1.00$ | 1594 | $77.13 \pm 1.30$ | 349 | $70.23 \pm 1.42$ | 341 | $75.64 \pm 1.33$ | 341 | $\mathbf{77.33 \pm 1.34}$ | 342 |
| 256 | $88.20 \pm 0.97$ | 1600 | $86.93 \pm 1.01$ | 1627 | $78.19 \pm 1.33$ | 340 | $75.61 \pm 1.42$ | 342 | $\mathbf{78.17 \pm 1.28}$ | 343 | $\mathbf{77.56 \pm 1.34}$ | 343 |
| SDPA | $88.75 \pm 1.00$ | 1532 | — | — | $\mathbf{78.06 \pm 1.33}$ | 344 | — | — | — | — | — | — |

We consider DeepSeek-R1-Distill-Qwen-7B (hereafter "DeepSeek 7B") (DeepSeek-AI, 2025) and Llama-3.1-8B-Instruct (hereafter "Llama 8B") (Meta AI, 2024). Table 2 shows accuracy on the GSM8K dataset as a mathematical reasoning benchmark (Cobbe et al., 2021; OpenAI, 2023). We evaluate the test split (1319 prompts) 3 times. Evaluation code uses straightforward answer parsing and borrows grading code from PRM800K (Lightman et al., 2023); for full prompting logistics, please see Supplementary Material. For both models, temperature $= 0.6$, top-p $= 0.95$, and repetition penalty $= 1.1$.

Chain-of-thought is built-in to DeepSeek 7B, leading to verbose output. Generation length is capped at 4096 tokens (excluding the prompt). Llama 8B is more succinct in responses, so generation length is capped at 1024 new tokens.

Table 2 provides the average number of tokens (prompt + answer) because the compute and memory access costs of SANTA are contextualized relative to the sequence length $n_k$. Following the analyses of Table 1, SANTA is far cheaper than SDPA provided $S << n_k$, incurring $S/n_k$ of SDPA's adds and memory accesses to $V$, and none of SDPA's multiplies.

Default SANTA exhibits respectable performance, but the $S^2$ANTA variants demonstrate superior performance across every sample budget $S$. For Llama-8B, $S^2$ANTA-ind and $S^2$ANTA-sys demonstrate similar performance to top-k for $k = S$ despite $S^2$ANTA requiring no multiplies. $S^2$ANTA implementations approach the accuracy of full SDPA (within 1%) while $S$ remains shorter than the sequence length $n_k$, indicating memory access savings and FLOP energy savings. In deployment settings where LLM users do not require maximum model performance, the sample budget $S$ can be tuned to save computation cost. For instance, $S^2$ANTA-sys achieves 76.42% accuracy at $S = 64$, which is about 19% of the average sequence length of 341 tokens. For a modest accuracy tradeoff compared to SDPA's 78.06%, $S^2$ANTA-sys in principle costs 19% of the additions, 19% of the $V$-matrix accesses, and *no multiplies*.

## 4.3 MMLU

Table 3: MMLU accuracy and average context length (prompt + answer). $k$: number of keys in top-$k$, $S$: SANTA sample budget. Accuracy shows 95% bootstrap confidence intervals.

| | DeepSeek-R1-Distill-Qwen-7B | | | | Llama-3.1-8B-Instruct | | | | | | | |
| | Top-$k$ | | SANTA | | Top-$k$ | | SANTA | | $S^2$ANTA-ind | | $S^2$ANTA-sys | |
| $k \mid S$ | Acc. (%) | Tok. | Acc. (%) | Tok. | Acc. (%) | Tok. | Acc. (%) | Tok. | Acc. (%) | Tok. | Acc. (%) | Tok. |
|---|---|---|---|---|---|---|---|---|---|---|---|---|
| 2 | $24.97 \pm 1.24$ | 3598 | $4.33 \pm 0.58$ | 4178 | $23.56 \pm 1.20$ | 588 | $24.89 \pm 1.27$ | 1144 | $24.28 \pm 1.21$ | 1141 | $24.52 \pm 1.22$ | 1141 |
| 4 | $38.93 \pm 1.32$ | 2236 | $23.10 \pm 1.20$ | 3666 | $28.37 \pm 1.27$ | 359 | $25.13 \pm 1.20$ | 961 | $24.65 \pm 1.26$ | 807 | $24.49 \pm 1.18$ | 774 |
| 8 | $55.08 \pm 1.47$ | 1659 | $44.55 \pm 1.47$ | 2080 | $39.54 \pm 1.43$ | 532 | $25.06 \pm 1.23$ | 284 | $25.63 \pm 1.23$ | 286 | $24.47 \pm 1.23$ | 292 |
| 16 | $59.07 \pm 1.36$ | 1301 | $58.81 \pm 1.42$ | 1506 | $45.26 \pm 1.37$ | 484 | $26.24 \pm 1.26$ | 313 | $32.46 \pm 1.32$ | 350 | $34.14 \pm 1.39$ | 353 |
| 32 | $60.77 \pm 1.40$ | 1173 | $61.05 \pm 1.45$ | 1228 | $49.49 \pm 1.44$ | 448 | $33.81 \pm 1.34$ | 354 | $43.78 \pm 1.50$ | 398 | $45.48 \pm 1.45$ | 403 |
| 64 | $62.03 \pm 1.38$ | 1103 | $60.59 \pm 1.43$ | 1104 | $48.33 \pm 1.37$ | 433 | $41.11 \pm 1.40$ | 397 | $\mathbf{49.25 \pm 1.48}$ | 415 | $\mathbf{48.70 \pm 1.39}$ | 418 |
| 128 | $61.68 \pm 1.49$ | 1061 | $\mathbf{62.38 \pm 1.32}$ | 1075 | $49.47 \pm 1.44$ | 418 | $47.46 \pm 1.45$ | 412 | $\mathbf{49.92 \pm 1.47}$ | 421 | $\mathbf{50.60 \pm 1.37}$ | 421 |
| 256 | $62.05 \pm 1.46$ | 1041 | $\mathbf{62.27 \pm 1.39}$ | 1074 | $49.75 \pm 1.43$ | 421 | $\mathbf{49.75 \pm 1.51}$ | 417 | $\mathbf{50.90 \pm 1.49}$ | 422 | $\mathbf{51.12 \pm 1.46}$ | 421 |
| SDPA | $\mathbf{63.16 \pm 1.45}$ | 1020 | — | — | $\mathbf{49.86 \pm 1.47}$ | 424 | — | — | — | — | — | — |

We evaluate on the MMLU benchmark (Hendrycks et al., 2020; Center for AI Safety, 2024), which tests factual recall and general reasoning. Each configuration in Table 3 is averaged over three runs on the validation set (1531 questions), using identical decoding parameters as GSM8K (Table 2).

MMLU benchmarking observes similar trends as GSM8K, with default SANTA showing reasonable performance, matching top-k for both DeepSeek and Llama models despite the non-straightforward comparison at $k = S$. SANTA virtually recovers baseline accuracy for both models at $S = 256$, which is importantly significantly shorter than the sequence length $n_k$.

For Llama-8B, $S^2$ANTA variants outperform both top-k and default SANTA across $S$ and $k$ budgets of $64 - 256$, recovering baseline SDPA accuracy (within 1%) with multiplier-free arithmetic and sparse memory access to rows of $V$.

## 4.4 Long context benchmarks

We evaluate SANTA on long-context tasks borrowed from RULER (Hsieh et al., 2024) with a prompt length of 8192 tokens. Tasks include frequent-word extraction (FWE), needle-in-a-haystack (NIAH),

Table 4: Accuracy of Llama 8B on 4 long context tasks with an 8k-token prompt. Tasks: frequent word extraction (FWE), needle-in-a-haystack (NIAH), single-hop QA ($QA_1$), multi-hop QA ($QA_2$). $k$: number of keys for top-k. $S$: SANTA sample budget. Accuracy shows 95% bootstrap CIs.

| Kernel | $k\|S$ | FWE | NIAH | $QA_1$ | $QA_2$ |
|---|---|---|---|---|---|
| top-$k$ | 64 | $93.07 \pm 1.20$ | $92.55 \pm 1.10$ | $65.60 \pm 4.10$ | $60.80 \pm 4.10$ |
| top-$k$ | 128 | $95.40 \pm 0.96$ | $93.60 \pm 1.15$ | $69.20 \pm 3.90$ | $62.20 \pm 4.50$ |
| **top-$k$** | **256** | **$97.27 \pm 0.80$** | **$94.55 \pm 1.05$** | **$70.60 \pm 3.90$** | **$64.20 \pm 4.10$** |
| SANTA | 64 | $92.53 \pm 1.27$ | $93.95 \pm 1.30$ | $64.60 \pm 4.20$ | $63.40 \pm 4.20$ |
| SANTA | 128 | $94.53 \pm 1.13$ | $91.45 \pm 1.38$ | $67.80 \pm 4.10$ | $63.40 \pm 4.30$ |
| SANTA | 256 | $96.20 \pm 0.93$ | $93.15 \pm 1.18$ | $68.00 \pm 4.10$ | $66.40 \pm 4.10$ |
| $S^2$ANTA-ind | 64 | $95.93 \pm 0.94$ | $94.35 \pm 1.15$ | $71.40 \pm 4.00$ | $65.40 \pm 4.20$ |
| $S^2$ANTA-ind | 128 | $97.73 \pm 0.80$ | $94.30 \pm 1.10$ | $70.80 \pm 4.20$ | $67.40 \pm 4.20$ |
| **$S^2$ANTA-ind** | **256** | **$98.33 \pm 0.60$** | **$94.55 \pm 1.05$** | **$71.20 \pm 4.00$** | **$67.00 \pm 4.20$** |
| $S^2$ANTA-sys | 64 | $96.27 \pm 0.97$ | $94.05 \pm 1.07$ | $71.80 \pm 3.90$ | $67.20 \pm 4.20$ |
| $S^2$ANTA-sys | 128 | $97.80 \pm 0.70$ | $94.15 \pm 1.02$ | $68.20 \pm 4.10$ | $66.80 \pm 4.00$ |
| **$S^2$ANTA-sys** | **256** | **$97.87 \pm 0.70$** | **$95.00 \pm 0.95$** | **$71.20 \pm 3.90$** | **$69.40 \pm 4.10$** |
| **SDPA (baseline)** | **—** | **$98.53 \pm 0.60$** | **$94.55 \pm 1.00$** | **$71.80 \pm 3.90$** | **$68.60 \pm 4.20$** |

single-hop question answering ($QA_1$ derived from SQuAD (Rajpurkar et al., 2018)), and multi-hop question answering ($QA_2$, derived from HotpotQA (Yang et al., 2018)). FWE prompts the model to extract the three most frequent words from the context. NIAH entails extracting three numeric needles from the prompt. Each entry of Table 4 shows accuracy averaged over 500 prompts.

In table 4, $S^2$ANTA consistently outperforms top-k on long contexts, particularly in multi-hop QA ($QA_2$) where unbiased sampling better captures distributed information ($\approx 5\%$ advantage for all $S = k$ settings). It is plausible that the benefits of $S^2$ANTA's unbiasedness is more apparent at long contexts, as information is spread across more keys and $S^2$ANTA is better able to capture heavy-tailed attention distributions.

$S^2$ANTA's benefits are most stark in this long-context regime. With a budget of $S = 256$ in an 8192-token sequence, **$S^2$ANTA uses just 3.125% of the value-stage additions and memory accesses compared to full attention** (following Table 1). It achieves this while using **no multiplications** and recovering **nearly all of the baseline SDPA accuracy** (within $\pm 1\%$) across every task.

### 4.5 FLOP SCALING

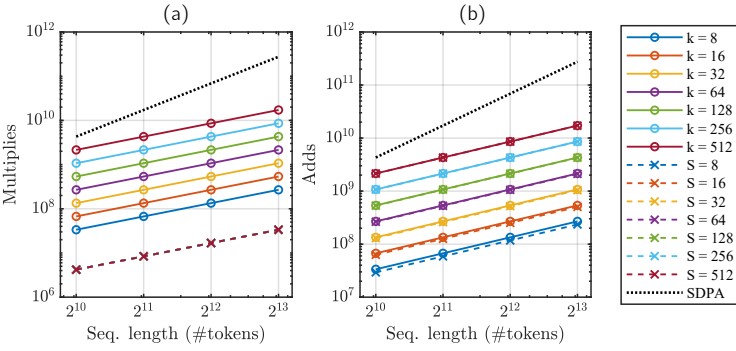

Figure 2: FLOP profile for the value stage computation. (a) Multiplications: SDPA scales quadratically with sequence length, top-$k$ scales linearly, and SANTA incurs effectively no multiplies. (b) Additions: SANTA and top-$k$ scale linearly and match closely.

We analyze ideal FLOP scaling in the value stage of Llama 8B (32 layers, $H=32$, $d_k=128$) during a single forward pass. Fig. 2 shows the number of multiplications and additions incurred by SDPA, top-$k$, and SANTA as sequence length increases. Multiplications (Fig. 2a) grow quadratically in SDPA. Top-$k$ reduces this to linear, while SANTA eliminates them entirely, aside from a per-head

normalization by $S$, which becomes a bitshift when $S$ is a power of 2. For fixed $S$ and $k$, additions (Fig. 2b) scale linearly for both SANTA and top-$k$, with nearly identical slopes. When $k \approx S$, the number of additions is comparable, but only SANTA eliminates all multiplications.

## 4.6 UNIQUE KEY-COUNT

Table 5: Unique-key count vs. $S$ (8192 tokens).

| $S$ | SANTA | $S^2$ANTA-ind | $S^2$ANTA-sys |
|-----|-------|---------------|---------------|
| 32  | 11.0  | 12.7          | 13.0          |
| 64  | 16.7  | 21.2          | 21.7          |
| 128 | 25.9  | 35.8          | 36.4          |
| 256 | 38.3  | 60.3          | 61.6          |

Since all versions of SANTA sample keys with replacement, the number of *unique* keys may be less than the sample budget $S$. In Table 5, we empirically measure the average number of unique keys sampled by the last query position for 8192-token single-hop QA prompts (normalized per head and averaged across model layers). For a given $S$, $S^2$ANTA variants achieve higher key diversity than default SANTA. Furthermore, we remark that the number of unique keys tends to be $<< S$, implying that *fewer than $S$* unique rows of $V$ require memory accesses - thus the $n_q S d_k$ memory reads estimate in Table 1 is a worst-case figure, and in practice, we access fewer unique keys.

## 4.7 EMPIRICAL VARIANCE

To further validate our results, we empirically measure the variance of SANTA, $S^2$ANTA-ind, and $S^2$ANTA-sys. We sweep the sample budget $S$ on 8k-token single-hop QA prompts, averaged across model layers and 30 prompts (normalized per head, see Appendix E). $S^2$ANTA variants exhibit significantly lower variance than SANTA. Though systematic sampling admits no tractable theoretical variance guarantees, it empirically demonstrates similar variance to independent stratified sampling (and requires 1 random number per query instead of $S$ random numbers). On a log-log scale, slopes for SANTA, $S^2$ANTA-ind, and $S^2$ANTA-sys are $-1.1$, $-1.27$, and $-1.29$, respectively. This perfectly reflects the $1/S$ scaling prescribed by theory (Prop 3.4).

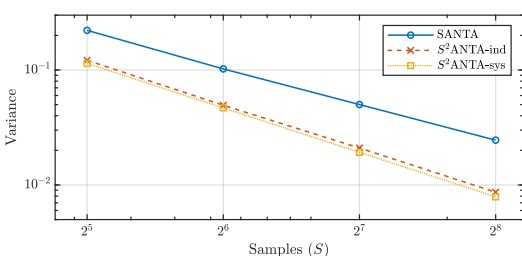

Figure 3: Empirical variance of SANTA on single-hop QA prompts (8k tokens).

## 5 CONCLUSION

SANTA is an unbiased estimator that replaces all multiplications in the value stage with additions, indexing, and a bit shift. It achieves accuracy competitive with top-$k$ and SDPA on GSM8K, MMLU, and long-context prompts, while eliminating a major source of compute. Theoretical analysis and FLOP profiling show that SANTA matches top-$k$ and significantly beats SDPA in memory access and addition cost, but incurs *no multiplications*.

SANTA is complementary and orthogonal to methods targeting $QK^\mathsf{T}$ or feedforward layers, including quantization, low-rank kernels, and pruning. Together, they point toward completely multiplier-free attention at inference.

We position SANTA as a forward-looking algorithmic contribution. While we make hardware-agnostic claims, we recognize that SANTA's sparse, data-dependent memory accesses are a departure from the dense matrix operations for which contemporary GPUs are optimized. However, inference on edge devices or alternative accelerators is increasingly viable. Architectures like BitNet already replace weight multipliers with cheaper operations like addition via ternary quantization. While these pipelines have removed many multiplies, they still compute $Q$, $K$, and $V$ matmuls. SANTA removes value-stage multiplies, charting a clear path toward fully multiplier-free transformers. SANTA's ability to recover high-quality outputs with a small, tunable sample budget with multiplier-free arithmetic makes it a compelling and practical solution for the next generation of energy-constrained AI hardware.

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

## A  VECTOR BERNSTEIN TAIL FOR SANTA

**Theorem A.1** (Vector Bernstein tail for SANTA). *Assume a uniform almost-sure bound $\|V_j - \mu_q\|_2 \le L$ for all $j$ (hence $\|X_s\|_2 \le L$). Let $v_q \triangleq \mathbb{E}\big[\|V_i - \mu_q\|_2^2\big] = \operatorname{tr}(\Sigma_q)$. Then for all $t > 0$,*

$$\Pr\Big(\big\|\widehat{V}_q - \mu_q\big\|_2 \ge t\Big) \ \le\ 2\exp\left(-\frac{S\,t^2}{2\,(v_q + Lt/3)}\right).$$

*Equivalently, with probability at least $1 - \delta$,*

$$\big\|\widehat{V}_q - \mu_q\big\|_2 \ \le\ \sqrt{\frac{2\,v_q\,\log(2/\delta)}{S}} \ + \ \frac{2L\,\log(2/\delta)}{3S}.$$

*Proof sketch.* This is a Hilbert-space (vector) Bernstein inequality: apply Bernstein to the mean of independent, mean-zero, $L$-bounded vectors with variance proxy $v_q$; see, e.g., Boucheron et al. (2013, Thm. 6.1) or Pinelis (1994). A coordinatewise Bernstein plus an $\varepsilon$-net on $S^{d_v-1}$ also yields the stated form (up to absolute constants) (Vershynin, 2018, Sec. 5.2). □

# B    VARIANCE REDUCTION FOR $S^2$ANTA-IND

**Theorem B.1** (Variance reduction for $S^2$ANTA-ind). *Let $\Sigma_q := \sum_j p_{qj}(V_j - \mu_q)(V_j - \mu_q)^\top$ and define within-stratum covariances*

$$\Sigma_q^{(m)} := \mathrm{Cov}\Big(V_{F_q^{-1}(T)} \,\Big|\, T \sim \mathrm{Unif}(I_m)\Big).$$

*Then*

$$\mathrm{Cov}\big(\widehat{V}_q^{\mathrm{ind}}\big) = \frac{1}{S^2} \sum_{m=0}^{S-1} \Sigma_q^{(m)} \ \preceq_{\mathrm{Loewner}} \ \frac{1}{S} \Sigma_q = \mathrm{Cov}(\widehat{V}_q),$$

*with strict improvement unless all stratum means coincide. See Lohr (2010, Ch. 3) and Cochran (1977, Ch. 8).*

*Proof.* Independence across strata gives $\mathrm{Cov}(\widehat{V}_q^{\mathrm{ind}}) = \frac{1}{S^2} \sum_m \mathrm{Cov}\Big(V_{F_q^{-1}(T)} \mid T \in I_m\Big)$. By the law of total covariance with $T \sim \mathrm{Unif}(0,1)$ and partition $\{I_m\}$,

$$\Sigma_q = \mathbb{E}\Big[\mathrm{Cov}(V_{F_q^{-1}(T)} \mid T \in I_m)\Big] + \mathrm{Cov}\Big(\mathbb{E}[V_{F_q^{-1}(T)} \mid T \in I_m]\Big) \succeq_{\mathrm{Loewner}} \mathbb{E}\Big[\mathrm{Cov}(V_{F_q^{-1}(T)} \mid T \in I_m)\Big],$$

and averaging then dividing by $S$ yields the claim. Strictness holds unless the stratum means are all equal. □

**Corollary B.2** (MSE ordering). *For any $q$, $\mathbb{E}\Big[\|\widehat{V}_q^{\mathrm{ind}} - \mu_q\|_2^2\Big] \leq \mathbb{E}\Big[\|\widehat{V}_q - \mu_q\|_2^2\Big] = \frac{1}{S} \mathrm{tr}(\Sigma_q).$*

# C    ABLATION STUDY: ONE-HOT ATTENTION SAMPLES

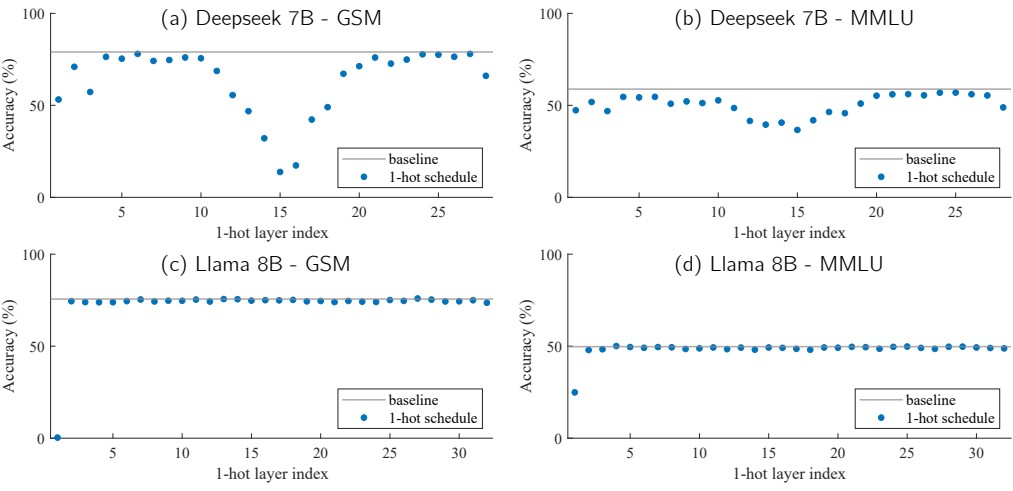

Figure 4: Ablations with one-hot stochastic attention. Baseline scores employ a fixed sampling budget of 16 and 256 for DeepSeek 7B and Llama 8B models, respectively. The horizontal axis shows the index of the model layer which is reduced to stochastic hard attention.

This section probes a fundamental question: how much does each attention layer matter? We find that one-hot stochastic attention, a limiting case of SANTA with $S = 1$, acts as a surprisingly sharp diagnostic. It reveals which transformer layers are robust to severe approximation and which are not, offering a practical lens into where attention precision matters most.

We perform these ablations by setting $S = 1$ in a single layer while keeping all other layers at fixed sample budgets: $S = 16$ for DeepSeek 7B (28 layers) and $S = 256$ for Llama 8B (32 layers). In this setting, each query attends to only one randomly sampled key, making that layer a hard attention bottleneck.

Models are evaluated on GSM8K and MMLU with the same protocol as Tables 2 and 3. Fig. 4 shows the effect of this localized bottleneck on accuracy. Some layers exhibit almost no degradation when ablated in this way, while others collapse entirely, most notably middle layers in DeepSeek 7B and the first layer in Llama 8B.

In DeepSeek 7B (Fig. 4a,b), layers 12–18 are highly sensitive to approximation, while early and final layers are much more tolerant. In contrast, Llama 8B (Fig. 4c,d) exhibits extreme sensitivity in just the first layer. These patterns suggest that attention importance is neither uniform nor trivially architectural.

We observe similar qualitative trends regardless of the baseline sample budget; the $S = 16$ and $S = 256$ settings are arbitrary. The early-layer sensitivity in Llama 8B is consistent with prior findings on high-entropy attention in shallow layers (Clark et al., 2019), and with prior work that avoids approximating early layers (Tang et al., 2024; Sun et al., 2024). The middle-layer fragility in DeepSeek 7B remains an open and intriguing observation.

## D  LAYER-WISE SAMPLE BUDGET THROUGH RL

---

**Algorithm 1** Layer-wise sample-budget optimization via REINFORCE

---

**Require:** initial logits $\boldsymbol{\theta} \in \mathbb{R}^{28}$ ($\theta_i = 0$),  total budget $N{=}224$, learning-rate $\alpha{=}0.02$
 1: **while** training **do**
 2:  **Workers (in parallel):**
 3:  **for** $e = 1, \ldots, E$ **do**                    $\triangleright$ $E{=}10$ episodes / worker run
 4:   Compute $p_i \leftarrow \exp(\theta_i)/\sum_j \exp(\theta_j)$
 5:   Sample schedule $\mathbf{S} \sim \text{Multinomial}(N, \mathbf{p})$
 6:   Evaluate 16 GSM8K questions using schedule $\mathbf{s}$; obtain reward $r \in [0, 1]$
 7:   Append $(\mathbf{s}, r)$ to shared folder
 8:  **end for**

 9:  **Aggregator:**
10:  Collect all new $(\mathbf{s}^{(k)}, r^{(k)})$ tuples              $\triangleright$ $k = 1 \ldots K$
11:  Compute baseline $b \leftarrow \frac{1}{K} \sum_k r^{(k)}$
12:  **Gradient:** $\mathbf{g} \leftarrow \dfrac{1}{K} \sum_k (r^{(k)} - b)(\mathbf{s}^{(k)} - N\mathbf{p})$
13:  Update logits $\boldsymbol{\theta} \leftarrow \boldsymbol{\theta} + \alpha\,\mathbf{g}$
14:  Write back updated $\boldsymbol{\theta}$
15: **end while**

---

We now consider a global sample budget shared across all 28 transformer layers of DeepSeek 7B, allowing each layer to have a distinct sample count. We optimize this allocation using a REINFORCE-style algorithm (Williams, 1992), detailed in Algorithm 1.

We train the policy on the GSM8K training set. The policy logits $\boldsymbol{\theta} \in \mathbb{R}^{28}$ define a softmax distribution over layers. At each iteration, we sample schedules using a multinomial draw over this distribution, with a total budget of 224 samples (i.e., $8 \times 28$). Each episode evaluates 16 questions; the reward is the fraction answered correctly. We intentionally starve the baseline with $S{=}8$ samples per layer to create room for the learned schedule to improve.

Fig. 5a shows the final learned schedule after 501 iterations. The allocation aligns strikingly with the ablation trends from Fig. 4: more samples are assigned to the first and middle layers, where one-hot attention was most detrimental. That REIN-FORCE independently rediscovers this structure from only

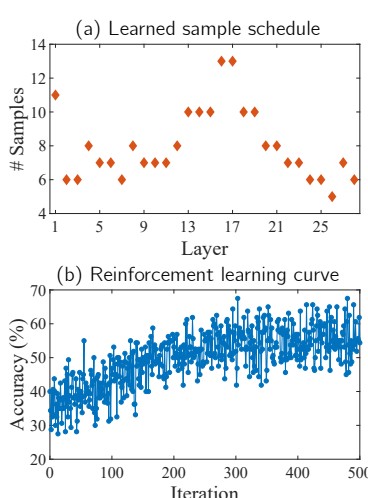

Figure 5: (a) Learned schedule with a total budget of 224 samples across all 28 transformer blocks (DeepSeek 7B). (b) RL iterations over time.

reward signals confirms that these patterns are not artifacts of the ablation method but instead are intrinsic to the models.

Fig. 5b shows the policy's baseline accuracy over time. While noisy, the curve plateaus as the schedule stabilizes. This experiment is not tuned for peak accuracy, but demonstrates that adaptive per-layer budgets meaningfully outperform fixed ones.

Table 6: RL sample-schedule performance. Accuracy shows 95% bootstrap confidence intervals.

| Schedule | GSM8K accuracy (%) | MMLU accuracy (%) |
|---|---|---|
| Baseline (8 samples) | $52.46 \pm 1.47$ | $44.55 \pm 1.47$ |
| Learned schedule | $\mathbf{66.06 \pm 1.47}$ | $\mathbf{48.86 \pm 1.53}$ |

Table 6 shows final scores using the learned schedule. With the same total compute, the RL schedule yields a 13.6% boost on GSM8K and a 4.3% gain on MMLU over the fixed-budget baseline. These results confirm that per-layer sample allocation can be learned directly from task reward and matters substantially.

## E  IMPLEMENTATION & MEASUREMENT PROTOCOL FOR SANTA VARIANTS

**Setting.** For a single layer/head at the last prefill position $q^\star$, let the post-softmax attention over $K$ keys be $p_j \geq 0$ with $\sum_{j=1}^K p_j = 1$, and let $V_j \in \mathbb{R}^D$ denote the corresponding value vectors. The dense-attention mean is

$$\mu = \sum_{j=1}^K p_j\, V_j \;\in\; \mathbb{R}^D.$$

**SANTA estimator.** Given a per-head sampling budget $S$, all SANTA variants estimate $\mu$ with

$$\widehat{V} = \frac{1}{S}\sum_{s=1}^S V_{J_s},$$

where the key indices $\{J_s\}$ are drawn by one of three schemes: (i) multinomial; (ii) independent equal-mass stratified; or (iii) systematic (random-start, fixed-stride). These are the same variants used in the main paper; the code paths that implement them also record the per-head statistics described below at $q^\star$.

### E.1  WHAT WE MEASURE (PER HEAD AT $q^\star$)

**(1) Unique keys.** The number of distinct keys touched by the sampler,

$$U := \big|\{J_1, \ldots, J_S\}\big|,$$

upper-bounded by $S$ (can be $< S$ if high-probability keys repeat). This is a proxy for how many memory locations each head actually reads.

**(2) Variance trace of the estimator.** We report the trace of the covariance (expected squared $\ell_2$ error) of $\widehat{V}$,

$$\mathrm{VarTrace}(\widehat{V}) := \mathbb{E}\big[\|\widehat{V} - \mu\|_2^2\big] = \mathrm{tr}\Big(\mathrm{Cov}(\widehat{V})\Big),$$

specialized to each sampling scheme:

- **Multinomial (closed form).** Let $\Sigma := \sum_{j=1}^K p_j\,(V_j - \mu)(V_j - \mu)^\top$ so that $\mathrm{tr}(\Sigma) = \sum_j p_j\|V_j\|_2^2 - \|\mu\|_2^2$. Then

$$\mathrm{VarTrace}_{\mathrm{multi}}(\widehat{V}) = \frac{1}{S}\,\mathrm{tr}(\Sigma).$$

This is computed directly from $\{p_j, V_j\}$ for the last-query row.

- **Independent equal-mass stratified (closed form).** Partition $[0, 1)$ into $S$ equal strata; for each stratum $m$, form the within-stratum discrete distribution induced by $p$ and let $\Sigma_m$ be the corresponding value covariance. Because draws across strata are independent,

$$\text{VarTrace}_{\text{strat}}(\widehat{V}) \;=\; \frac{1}{S^2} \sum_{m=0}^{S-1} \text{tr}(\Sigma_m).$$

We evaluate this design-based expression exactly for the discrete case at $q^\star$.

- **Systematic (replicate-based estimate).** There is no general closed form that holds uniformly for systematic sampling, so we estimate variance by repeated random starts. With $R \geq 2$ independent offsets, we compute $\widehat{V}^{(r)}$ for each replicate, then use

$$\widehat{\text{VarTrace}}_{\text{sys}}(\widehat{V}) \;=\; \frac{1}{R-1} \sum_{r=1}^{R} \big\| \widehat{V}^{(r)} - \overline{V} \big\|_2^2, \quad \overline{V} = \frac{1}{R} \sum_{r=1}^{R} \widehat{V}^{(r)}.$$

This estimator is applied per head at $q^\star$.

## E.2 How we aggregate and report

For each prompt, we compute $U$ and the variance-trace scalar per head at $q^\star$, then average over heads within a layer, average over layers, and finally average over prompts to obtain the quantities reported in the variance plots:

$$\overline{U}(S, \text{variant}) \quad \text{and} \quad \overline{T}(S, \text{variant}) \;=\; \overline{\text{VarTrace}}(\widehat{V}).$$

This yields the curves summarized in the main text as a function of the budget $S$ and the SANTA variant.

**Protocol summary.** All measurements use evaluation-mode forward passes, prefill-only, and record statistics *at the last token* per head; no additional dense attention pass is performed to form empirical errors. For multinomial and stratified we use the exact formulas above; for systematic we use the replicate-based estimator.

# F LLM usage statement

LLMs were used to polish the presentation and writing of this contribution.

# G Reproducibility statement

Computation for GSM8K and MMLU prompts uses 24GB NVIDIA GPUs from the Ampere generation (L40, A10, 3090), while long-context prompts employ NVIDIA RTX A6000 GPUs. Models and datasets are open-source and properly credited. Codes for our methods are provided at `https://anonymous.4open.science/r/SANTA-718E`.

