# Supplementary Material: Towards Multiplier-Free Transformers with Stochastic Attention

## 1 Prompting, Parsing, and Grading

**Scope.** Four model–dataset pairs are evaluated with left-padded batches of 8–16 prompts:

- DeepSeek-R1-Distill-Qwen-7B + GSM8K
- DeepSeek-R1-Distill-Qwen-7B + MMLU
- Llama-3.1-8B-Instruct + GSM8K
- Llama-3.1-8B-Instruct + MMLU

For GSM8K and MMLU, generation uses `temperature=0.6`, `top_p=0.95`, and `repetition_penalty=1.1`. For long-context tasks, decoding is greedy. We remark that baseline attention implementations and SANTA are compared in the scope of a single model-dataset pair, thus parameter choices do not affect conclusions so long as they are consistent across attention implementations.

**DeepSeek 7B + GSM8K**

| Aspect | Procedure |
| --- | --- |
| Prompt | Question, blank line, then `Please reason step by step, and put your final numeric answer within \boxed{}`. |
| Answer extraction | Perform a brace-balanced search for the last occurrence of `\boxed{...}` (robust to nested braces). If none is found, fall back to the final ~200 characters of the model output. |
| Grading | Numeric equality judged by the MIT-licensed PRM800K grader; every prompt is counted, and blank or unparsable predictions score 0. |

Table S1: Prompting and grading for DeepSeek 7B on GSM8K.

**DeepSeek 7B + MMLU**

| Aspect | Procedure |
| --- | --- |
| Prompt | Question, four labeled choices, then `Answer briefly. Put your final answer as a single letter inside \boxed{}`. |
| Answer extraction | Brace-balanced search for the last occurrence of `\boxed{...}`; if none is found, inspect the final ~300 characters of the output. Scan this region backwards and take the first occurrence of A–D (case-insensitive), then convert it to uppercase. |
| Grading | The predicted letter is compared with the gold key (case-insensitive). If no valid letter is found—or the output is blank—the item scores 0; otherwise it scores 1 when the letters match. |

Table S2: Prompting and grading for DeepSeek 7B on MMLU.

**Llama 8B + GSM8K**

| Aspect | Procedure |
| --- | --- |
| Prompt | Meta chat template. *System*: "You are a precise mathematician. Explain your reasoning step by step, then output ONLY the final number on a new line." *User*: the GSM8K question. |
| Answer extraction | Inspect the model reply in order: (i) the last non-empty line of the response; (ii) if that fails, the final 25 whitespace-separated tokens. From the selected chunk, take the first integer-looking token (commas allowed, optional leading minus). |
| Grading | Numeric equality judged by the MIT-licensed PRM800K grader; every question is counted, and blank or unparsable answers score 0. |

Table S3: Prompting and grading for Llama 8B on GSM8K.

**Llama 8B + MMLU**

| Aspect | Procedure |
| --- | --- |
| Prompt | Meta chat template. *System*: "You are a knowledgeable and concise subject-matter expert. Work through the problem step by step. Finally, on a new line, output ONLY the single capital letter (A, B, C, or D) that corresponds to the correct choice." *User*: the question followed by the four labeled choices A–D. |
| Answer extraction | Take the substring after the last newline and scan it left-to-right for the first capital A–D (case-insensitive). If none is found, inspect the final 10 whitespace-separated tokens of the whole response, scanning them right-to-left for A–D. |
| Grading | Predicted letter (upper-cased) is compared with the gold key. Every question is counted; if no valid letter is found or it does not match the gold answer, the item scores 0. |

Table S4: Prompting and grading for Llama 8B on MMLU.

**Llama 8B + RULER prompts**

| Aspect | Procedure |
| --- | --- |
| Prompt | Prompts are generated for the following RULER sub-tasks with a length of 8192 tokens: fwe, niah_multivalue, qa_1, and qa_2. Greedy decoding is used. The model generates a maximum of 64 new tokens. |
| Grading | fwe: partial credit = (# of gold words present in the model output) / 3 (case-insensitive set match). niah_multivalue: take the first four unique integers from the model output; partial credit = (# of these that appear in the gold set) / 4. qa_1 and qa_2: correct if the gold answer string appears as a case-insensitive substring of the model output. |

Table S5: Prompting and grading for Llama 8B on RULER prompts.

**Licenses.**

- **Models.** deepseek-ai/DeepSeek-R1-Distill-Qwen-7B (MIT) and meta-llama/Meta-Llama-3.1-8B-Instruct (Llama 3.1 license) on Hugging Face.
- **Datasets.** GSM8K openai/gsm8k (MIT) and MMLU cais/mmlu (MIT) on Hugging Face. RULER prompts (Apache) — `https://github.com/nvidia/ruler`
- **Grader.** PRM800K numeric grader (MIT) — `https://github.com/openai/prm800k`.

## 2 IMPLEMENTATION NOTES

The SANTA and top-$k$ routines are concise, self-contained Python scripts meant as reference implementations. They run under stock PyTorch and make no attempt at kernel fusion or memory tuning.

### 2.1 STOCHASTIC ADDITIVE NO-MULT ATTENTION (SANTA)

**Logical flow.** For each attention head the script

 (i) reshapes batch, head, and query axes into an $(n_q \times n_k)$ matrix of logits;

 (ii) draws $S$ categorical samples from every query's softmax distribution;

(iii) gathers the $S$ selected rows of $V \in \mathbb{R}^{n_k \times d_k}$;

(iv) accumulates the samples and divides by $S$ to yield an unbiased estimate of $AV$.

The only arithmetic after softmax is a vector addition per sample and one normalizing divide (a bit-shift if $S$ is a power of two).

**Two-way tiling.** A direct call to `torch.multinomial` or `torch.gather` allocates large temporaries: $n_q \times n_k$ and $n_q \times S \times d_k$, respectively. For long sequences these buffers exceed GPU memory, so the script processes

- **row tiles** of $T_r = 1024$ queries, and
- **sample tiles** of $T_s = 64$ samples inside each row tile.

Row-wise tiling spawns several small CUDA kernels and adds one per-tile normalization step. These are implementation-specific details that do not change the algorithm's mathematics.

### 2.2 TOP-$k$

Top-k is implemented as follows:

 (i) compute the full logits $QK^\mathsf{T} \in \mathbb{R}^{n_q \times n_k}$;

 (ii) retain the $k$ largest entries of each row, setting the rest to $-\infty$;

(iii) apply softmax and perform a dense matrix multiply with $V$.

Although a truly sparse $AV$ multiply is possible, the reference code performs the full product (many zeros included) to keep the implementation straightforward.

### 2.3 SCOPE AND LIMITATIONS

- Pure PyTorch. No Triton or custom CUDA.
- Code prioritises clarity over speed, and we make no latency or VRAM claims.
- **FLOP accounting.** Table 1 is a pen-and-paper tally; Fig. 2 derives FLOPs from tensor shapes in Llama-8B. We count only algorithm-intrinsic ops, omitting implementation artifacts such as SANTA's per-tile normalizing divides (which are negligible) and top-$k$ multiplies on zeroed logits. The exclusion of implementation-dependent operations does not alter any conclusion.

The scripts are reference implementations. Performance-tuned versions for GPU, CPU, or edge devices are future work.

Kubernetes (k8s) is an orchestration layer that runs containerized tasks (called "pods") across a cluster of machines while handling restarts and scaling automatically.

## 3 DISTRIBUTED REINFORCEMENT LEARNING ON KUBERNETES

**Topology.** All pods mount a single read–write volume that serves as the rendezvous point.

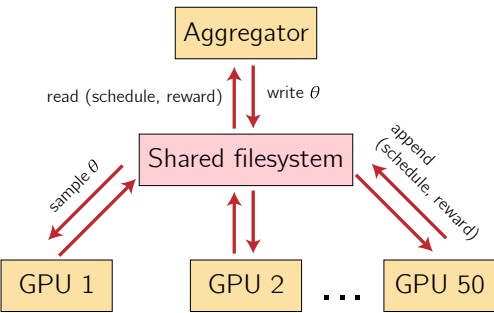

Figure S1: Kubernetes deployment schematic.

- **Aggregator pod.** Holds the current policy and records a running history of baselines and gradients.

- **Worker pods.** Operate in a tight loop: *(i)* read the latest policy; *(ii)* sample a 28-dimensional schedule; *(iii)* evaluate ten batches of 16 questions; *(iv)* append a small JSON-lines result file to a designated "inbox" directory on the volume.

**Aggregator loop.** Whenever at least one new result file appears, the aggregator moves all current files into a private folder for the next iteration, computes the REINFORCE update and writes a new policy snapshot. It never pauses or signals the workers; it simply processes whatever has arrived. In practice, workers are unlikely to finish evaluation simultaneously, thus nearly every policy update comes from one worker output ($10 \times 16$ questions). If two land together, the aggregator just uses a double-sized batch, which lowers the gradient's variance.

**Asynchrony and fault-tolerance.**

- Workers can be terminated at any time; partial output vanishes harmlessly.

- Workers always proceed with the newest policy that has reached disk; no locking is needed.

- If the aggregator restarts it scans the history directory, resumes from the last completed iteration, and continues.

**Scaling and resources.** The worker Deployment's replica count can scaled up or down at any time during learning. For the 501 iterations in Fig. 5 we ran 50 GPU workers on a mix of RTX 3090, L4, and A10 cards (24 GB Ampere generation cards) for roughly 20 h, totaling $\approx 1000$ GPU-hours. The aggregator requests just one CPU core and a few hundred MiB of RAM.

**Terminology.** We keep the names aggregator and worker to highlight the simple "parameter-server" flavor of the design, but these roles align exactly common learner/actor split in distributed reinforcement learning.