# OpenReview forum: "Towards Multiplier-Free Transformers with Stochastic Attention"
_ICLR.cc/2026/Conference — Submitted to ICLR 2026_

### Official Review · Reviewer_2czp · 2025-10-30

**Soundness:** 2
**Presentation:** 3
**Contribution:** 2
**Rating:** 4
**Confidence:** 3

**Summary:**

This paper introduces SANTA, an additive attention variant that serves as a drop-in replacement for the standard attention mechanism. SANTA approximates the attention output by sampling value vectors according to the post-softmax distribution and then computing their mean. The authors prove that this method provides an unbiased estimate and further propose S²ANTA, which leverages stratified sampling to reduce variance. In comparison to top-k attention, the authors demonstrate that SANTA is more power-efficient by completely eliminating multiplications in the value stage. The experiments show that SANTA achieves promising results, with performance comparable or even superior to the top-k method across various benchmarks.

**Strengths:**

1. This paper is well-written and easy to follow.
2. To the best of my knowledge, the method is novel.
3. SANTA effectively eliminates a key computational bottleneck while maintaining competitive model accuracy.
4. The finding that a simple, unbiased averaging of values can achieve performance comparable to the strong top-k baseline is both surprising and compelling.

**Weaknesses:**

1.  While the authors argue SANTA is more power-efficient by eliminating multiplications, they dismiss sampling costs as "lightweight relative to the V matrix multiply" (§3.4) without providing empirical measurements. The computational cost of sampling from categorical distributions over long sequences should be benchmarked against top-k's partial sorting overhead to substantiate the efficiency claims.
2. Although the authors claim the method is suitable for edge devices, the paper provides no actual hardware deployment experiments, nor does it discuss the implementation on edge devices.

**Questions:**

1. How does SANTA perform on actual edge hardware? While top-k can be fused with online softmax via heap-based algorithms with efficient rescaling, can SANTA similarly avoid full softmax I/O? Is sampling truly faster than partial sorting on resource-constrained devices?
2. What is the fundamental advantage of weighted Monte Carlo sampling over simply averaging the top-k values with uniform weights (also multiplication-free)? This ablation would clarify whether the stochastic framework provides benefits beyond just selecting a sparse subset.

Would raise my score if authors solve my questions.

---

> ### Author Response · Authors · 2025-12-03
> **Response to Reviewer 2czp**
>
> We thank Reviewer 2czp for the careful reading and positive assessment of the writing, novelty, and performance of SANTA relative to top-$k$ attention. We are especially encouraged that the reviewer found the empirical finding - that simple unbiased averaging of sampled values can match a strong top-$k$ baseline - to be compelling.
>
> *On power-efficiency claims and lack of hardware experiments.*
> We agree with the reviewer that our current claims about power-efficiency are based on operation counts and known energy models, not on direct hardware measurements. Our paper is best viewed as an *algorithmic* contribution: we show that one can entirely remove multiplications in the value stage of attention and still maintain competitive performance by using a Monte Carlo estimator with modest sample budgets $S$. Conceptually, this plays a similar role to the original scaled dot-product attention: **the mathematical formulation had to exist before optimized kernels such as FlashAttention or Flash Decoding could be developed**. SANTA is the corresponding mathematical step for multiplier-free value-stage attention: it shows that small samples of the attention distribution are sufficient to approximate the exact output, and it makes explicit the computational pattern (sparse indexed reads + additions) that future kernels or hardware can target.
>
> That said, we fully agree that:
> - Benchmarking categorical sampling versus top-$k$ partial sorting on real edge hardware, and
> - Providing end-to-end power and latency measurements
>
> would be important steps in translating the algorithmic idea into concrete systems gains.
>
> *On top-$k$ with uniform weights.*
> We appreciate the suggestion to compare SANTA against a baseline that selects the top-$k$ keys but applies uniform weights. This is a clean and informative ablation: it would help isolate the benefit of *importance-weighted* Monte Carlo sampling (using ${Categorical}(A_q)$) versus simply averaging a sparse subset. We agree this is a valuable experiment and will strongly consider including such a comparison in a future version.

---

### Official Review · Reviewer_X3US · 2025-10-30

**Soundness:** 4
**Presentation:** 3
**Contribution:** 3
**Rating:** 2
**Confidence:** 3

**Summary:**

This paper proposes SANTA (Stochastic Additive No-mulT Attention), a novel attention mechanism that aims to eliminate multiplication operations from Transformer inference.
Instead of computing the standard attention product
softmax(𝑄𝐾⊤)𝑉, SANTA samples keys from the attention distribution and approximates the output via Monte Carlo averaging. The resulting estimator 𝐴^𝑉A^V is shown to be unbiased, with variance decreasing proportionally to 1/𝑆, where 𝑆 is the number of samples.

An enhanced version, S²ANTA, applies stratified or systematic sampling to further reduce variance. Because the sampling process involves only additions, indexing, and bit shifts, the model achieves a completely multiplier-free inference pipeline when combined with low-bit quantized feed-forward layers (e.g., BitNet).

**Strengths:**

1. The stochastic sampling approach provides an unbiased, theoretically grounded estimator of the attention output.
2. Addresses a key limitation of efficient Transformers, the heavy reliance on multiplications in attention.

**Weaknesses:**

1. Lack of hardware validation: The claimed multiplier-free and energy-efficiency benefits are purely theoretical; no real measurements or FPGA/GPU latency/energy analysis are provided.
2. Potential implementation inefficiency: Random sampling and irregular memory access could make GPU execution slower than dense attention in practice.
3. Missing baselines: No comparison against kernelized linear attention or FlashAttention energy-profiling to substantiate the energy-efficiency claim.

**Questions:**

1. Can you provide any empirical runtime or energy measurements on real hardware (e.g., A100/H100 or FPGA) to substantiate the multiplier-free claim?
2. If the current hardware is not a good candidate for this algorithm, can the authors provide a clearer discussion on what type of hardware architecture would be suitable for such multiplier-free computation — for instance, what memory access patterns, parallelism model, or instruction primitives (e.g., stochastic sampling units or bit-level adders) would be required to efficiently support SANTA on future accelerators?

---

> ### Author Response · Authors · 2025-12-03
> **Response to Reviewer X3US**
>
> We thank Reviewer X3US for the detailed and constructive comments, and we are glad that the reviewer found the estimator theoretically grounded and the goal of eliminating multiplications in attention well-motivated.
>
> *On lack of hardware validation and potential inefficiency.*
> We agree that the present paper does not contain runtime or energy measurements on GPUs or FPGAs, and that sampling and irregular memory access patterns are crucial for practical efficiency. Our focus here is explicitly *algorithmic*: we formulate SANTA as an unbiased Monte Carlo estimator of
> $
> AV
> $
> that replaces the dense value-stage matmul by sampling, indexed gathers from $V$, additions, and a bit-shift. We show analytically that variance decreases as $1/S$ and empirically that modest $S << n_k$ preserves performance across benchmarks.
>
> Conceptually, this plays a similar role to the original scaled dot-product attention: **the mathematical formulation had to exist before optimized kernels such as FlashAttention or Flash Decoding could be developed**. SANTA is the corresponding mathematical step for multiplier-free value-stage attention: it shows that few samples $S$ of the attention distribution, relative to the sequence length $n_k$, are sufficient to approximate the exact output, and it makes explicit the computational pattern (sparse indexed reads + additions) that future kernels or hardware can target.
>
> We expect that:
> - On current GPUs, SANTA’s benefits would come primarily from reduced $V$ reads from the KV-cache (since only $S$ rows of $V$ are touched per query), and
> - On future addition-specialized or energy-constrained accelerators, eliminating value-stage multipliers could be more directly beneficial.
>
> For example, per Horowitz (2014), a 32-bit FP multiply costs roughly $3.7$\,pJ versus $0.9$\,pJ for an add; designs that can exploit this gap may find SANTA particularly attractive. **A full hardware characterization, however, is beyond the scope of this initial algorithmic paper.**
>
> *On baselines such as linear attention and FlashAttention.*
> We agree that comparisons to exact attention kernels like FlashAttention would be useful in a systems-oriented follow-up, especially from an energy perspective. For linear/Kernelized attention methods, we note that many such approaches are orthogonal to SANTA: whenever there is a softmax-$V$ stage (even over a reduced set of keys), SANTA can in principle be used to stochastically approximate that stage while removing value-stage multiplications. Exploring such compositions is an interesting direction for future work, but would require substantial additional engineering beyond the scope of this submission.

---

### Official Review · Reviewer_r9EV · 2025-10-31

**Soundness:** 3
**Presentation:** 3
**Contribution:** 2
**Rating:** 4
**Confidence:** 4

**Summary:**

The paper introduced Stochastic Additive No-multiplication Attention (SANTA), a drop-in, inference-time, replacement for the standard attention layer. SANTA approximates the attention operator without performing any score-value multiplications, thus offering a multiplier-free replacement. This is achieved through a Monte-Carlo estimation by sampling from the attention softmax distribution, thus only using additions, as well as a bit-shift operation instead of division (given the number of samples is a power of two).

**Strengths:**

The paper is easy to follow and clearly introduces the problem and its proposed attention method.

- Figure 1 is clear and is good representation of the algorithm.
- The mathematical notation is clearly introduced and the propositions and derivations are easy to follow.
- The authors showcase a good performance of SANTA compared to the sparse top-$k$ attention.

**Weaknesses:**

One of the main motivations of the paper seems to be getting rid of multipliers for transformer inference, complimenting techniques such as BitNet (that removes multipliers in matrix multiplications), and NoMAD-Attention, that removes multiplications in the $QK^T$ part of attention. The impact of this is not fully clear however, especially as, like the authors note, current hardware is indeed optimised for fast matrix multiplications. Unlike BitNet, SANTA is applied at inference time, and its implications to training are not explored. Furthermore, the authors hint at potential synergy of the method with BitNet-like architectures in order to fully get rid of multiplications in the forward calculation, but this is not backed by any experimental data (i.e., it is not clear if the method would work well with these architectures) — a demonstration of a fully multiplication-free transformer would benefit this argument.

- The underlying sampling idea is not new, and has been used in previous work to approximate attention (e.g., https://arxiv.org/abs/2410.16179)
- The paper does not provide any practical measurements on effective speed-ups that could be achieved. Although the algorithm might benefit future hardware to a greater extent, the decrease in FLOPs and memory transfers should still offer a benefit (and indeed top-$k$ techniques are used for this). Having a practical analysis could give a stronger case for using SANTA-like replacement of attention during inference.

**Questions:**

- The authors note that the number of unique keys accessed can be significantly fewer than the number of samples; it would be useful to get a sense of the implication of this to a practical speed-up.
- Although authors mention that sampling/sorting costs can be ignored, it would be helpful to get some sense of their relative cost vs. the other operations within the module. This could be especially helpful in order to understand the advantage of SANTA vs. top-$k$- as a drop-in replacement of attention during inference.
- Minor: It would be useful to also have perplexity results on a standard dataset (such as WikiText).

---

> ### Author Response · Authors · 2025-12-03
> **Response to Reviewer r9EV**
>
> We thank Reviewer r9EV for the thoughtful review. We are glad that the reviewer found the paper easy to follow, with clear notation and good performance relative to sparse top-$k$ attention.
>
> *On prior Monte Carlo attention (arXiv:2410.16179).*
> We agree that the underlying idea of using Monte Carlo estimators for attention is not new. In Section 2.4, we explicitly acknowledge that “Monte Carlo estimators have been proposed for efficient attention”, and we cite arXiv:2410.16179 as an important reference.
>
> Our contributions are complementary to this line of work. In particular:
> - SANTA is designed as a *post-softmax* Monte Carlo estimator that specifically removes multiplication by $V$ after the softmax step, turning the value-stage matmul into sampled gather-add + bit-shift.
> - We introduce S$^2$ANTA, showing how stratified and systematic sampling reduce variance in this attention setting.
> - Empirically, we systematically study how modest sample budgets $S \ll n_k$ preserve model quality while eliminating all value-stage multipliers.
>
> To our knowledge, this specific “no-multiplication in the value path” formulation plus variance-reduction analysis has not appeared before and should be of interest to the community.
>
> *On hardware speed-ups and implementation.*
> We agree that practical speed-ups and wall-clock measurements are important. However, the main contribution of this work is *algorithmic*: we show that standard attention
> $$
> AV, \quad A = {softmax}\left(\frac{QK^\top}{\sqrt{d_k}}\right)
> $$
> can be replaced by an unbiased Monte Carlo estimator that uses sampled gathers and additions instead of a dense value-stage matmul, with acceptable accuracy for modest $S$.
>
> Conceptually, this plays a similar role to the original scaled dot-product attention: **the mathematical formulation had to exist before optimized kernels such as FlashAttention or Flash Decoding could be developed**. SANTA is the corresponding mathematical step for multiplier-free value-stage attention: it shows that small samples of the attention distribution are sufficient to approximate the exact output, and it makes explicit the computational pattern (sparse indexed reads + additions) that future kernels or hardware can target.
>
> That said, we agree with the reviewer that:
> - Benchmarking sampling overhead versus top-$k$ partial sorting on current GPUs, and
> - Providing end-to-end wall-clock and energy numbers for SANTA kernels
>
> would significantly strengthen the case. These are natural directions for follow-up work.
>
> *On perplexity on standard datasets.*
> We appreciate the suggestion to report perplexity on a standard benchmark such as WikiText. This is a straightforward addition to our experimental protocol, and we will strongly consider including such results in a revised version.

---

### Official Review · Reviewer_51as · 2025-11-13

**Soundness:** 1
**Presentation:** 1
**Contribution:** 1
**Rating:** 0
**Confidence:** 2

**Summary:**

This paper sets out to remove the need for multiplication in attention networks by using what it calls stochastic attention.  The

**Strengths:**

The paper identifies a worthy goal: Reducing computational cost of neural networks that employ attention would be helpful.

**Weaknesses:**

While the introduction and related work sections (1 and 2) are readable, the main contribution section (3) was not in this reviewer's estimation actually human readable.  The paper says "LLMs were used to polish the presentation and writing of this contribution" on lines 843-844 but it's not clear what parts of the paper that applies to.  If this paper was written in a language other than English, then passed to an LLM to translate, I think the LLM did a poor job on the main technical part.

The results section appears to only compare against sparsity (top-k).  It is unclear why no other quantization approaches are compared against.

Regarding Remark 3.2 that if S = $2^m$ then division can be implemented as a bit shift, I am concerned what this paper is proposing in Equation 2 is to replace multiplication of A times B by summing B copies of A.   That hardly seems like the right way to achieve efficiency.

**Questions:**

Are you just proposing to implement multiplication by summing a bunch of copies of the multiplicand by the multiplier?  (That is what Equation 2 seems to be doing to me.)

I'm not sure what it means to "treat each row $A_q$ as a categorical distribution over keys and sample $S$ values i.i.d. from it".   What is $S$ and how is it set or selected?  What does it mean to sample a row of a matrix?  What is meant by categorical distribution?  How does any of this approximate a multiplication?

According to what distribution is the sampling of one-hot rows governed in the example in Equation 2?

For the "illustrative example" in Equation 2, what is the corresponding value of A?

It is unclear how $V_i$ is obtained in Equation 3.  What does it mean to stack rows from V in forming $V_i$?

What is "Categorical($A_q$)" on line 158?  What is $V_i_{q,s}$ in the equation on Lines 159-160?

---

> ### Author Response · Authors · 2025-12-03
> **Response to Reviewer 51as**
>
> Unfortunately, this review is based on several egregious factual errors and basic conceptual misunderstandings of our work. These misunderstandings appear to be the main reason for the strong reject recommendation, despite the reviewer explicitly choosing confidence level 2, which states that it is quite likely that the central parts of the submission were not understood. Given this, we respectfully request that the Area Chair disregard this review.
>
> At a high level, SANTA is a standard Monte Carlo estimator of the usual attention output
> $$
> AV, \quad A = {softmax}\left(\frac{QK^\top}{\sqrt{d_k}}\right).
> $$
> For each query $q$, the row $A_q$ is the usual attention weight vector over *key indices* $j \in \{1,\dots,n_k\}$ (which also index the rows of $V$). We sample $S$ indices
> $$
> i_{q,1},\dots,i_{q,S} \sim {Categorical}(A_q),
> $$
> gather the corresponding value rows $V_{i_{q,s}}$, and average:
> $$
> \hat V_q = \frac{1}{S}\sum_{s=1}^S V_{i_{q,s}}.
> $$
> A simple calculation gives
> $$
> {E}[\hat V_q] = \sum_{j=1}^{n_k} A_{qj} V_j = (AV)_q,
> $$
> so $\hat V_q$ is an unbiased Monte Carlo estimator of the exact attention output. **We therefore do *not* “implement multiplication by summing a bunch of copies of the multiplicand by the multiplier”** as the review incorrectly claims; we approximate the matrix product $AV$ via sampling, indexing, and addition, exactly as described and proved in Section 3.
>
> The one-hot matrices in Eq. (2) are explicitly introduced as a pedagogical device to illustrate that a one-hot attention matrix simply acts as an indexing operator into $V$. In the text we state that these one-hot matrices are not materialized and are used only to explain how sampling from ${Categorical}(A_q)$ corresponds to gathering rows from $V$. Reading Eq. (2) as a proposal to “sum copies” of a vector is a misreading of both the surrounding explanation and the formal propositions. Combined with the stated low confidence, this reinforces our view that this review does not provide a reliable assessment of the paper.
>
> Several of the reviewer’s questions (e.g., “What is meant by categorical distribution?”, “What does it mean to sample a row of a matrix?”) concern basic and explicitly defined concepts: $A_q$ is the standard attention row, ${Categorical}(A_q)$ is the **textbook discrete distribution** over key indices with probabilities $A_{qj}$, and $V_{i_{q,s}}$ is the row of $V$ at the sampled index $i_{q,s}$. These notions are standard in attention mechanisms and Monte Carlo methods and are spelled out in Definition 3.1 and the surrounding text. Taken together with the low confidence, this suggests non-engagement with the core construction rather than a substantive flaw in the method.

---

### Meta-Review · Area_Chair_jM9U · 2026-01-03

**Summary:**

This paper proposes SANTA, a stochastic Monte Carlo estimator that replaces the value stage matrix multiplication in attention with sampled gather-add operations, aiming to enable multiplier-free Transformer inference. Reviewers acknowledged the theoretical correctness of the estimator and the motivation to reduce multiplications, but raised significant concerns regarding practical performance improvements, lack of hardware validation, and insufficient empirical evidence supporting efficiency and energy claims.

**Reviewer Concerns:**

The rebuttal successfully clarified misunderstandings in reviews and strengthened the theoretical soundness of SANTA as an unbiased estimator of standard attention. However, the major concerns from multiple reviewers remain unresolved. The paper makes strong claims about multiplier-free and energy-efficient inference without providing any runtime, latency, or energy measurements on real hardware. Practical efficiency is therefore speculative, especially given the irregular memory access and sampling overhead. Additionally, comparisons are limited primarily to top-k attention, with missing baselines such as FlashAttention or kernelized attention variants, and no ablations studying the benefit of stochastic sampling over simpler multiplier-free alternatives (e.g., uniform top-k averaging). Overall, the contribution is primarily algorithmic and conceptual, but the empirical evidence is insufficient to support its claimed practical impact at ICLR.

**Reviewer Scores:**

Reviewer 51as: Likely increased from 0 due to clarification of misunderstandings but unlikely changed to positive.

Reviewer r9EV: Likely unchanged (4), remaining borderline due to lack of hardware validation.

Reviewer X3US: Likely unchanged (2), with concerns about speculative efficiency claims.

Reviewer 2czp: Likely unchanged (4), due to missing empirical validation.

---

### Decision · Program_Chairs · 2026-01-26

Reject